# The Effect of Novel Complex Treatment of Annealing and Sandblasting on the Microstructure and Performance of Welded TA1 Titanium Plate

**DOI:** 10.3390/ma16062149

**Published:** 2023-03-07

**Authors:** Yanbin Xu, Dayue Wang, Mingyen Li, Jing Hu, Xulong An, Wei Wei

**Affiliations:** 1Jiangsu Key Laboratory of Materials Surface Science and Technology, Huaide College, Changzhou University, Changzhou 213164, China; 2National Experimental Demonstration Center for Materials Science and Engineering, Changzhou University, Changzhou 213164, China; 3Changzhou Sinosteel Precision Forging Materials Co., Ltd., Changzhou 213150, China

**Keywords:** TA1 titanium plate, weld, heat treatment, electrolytic copper foil, cathode roller

## Abstract

The welding titanium cathode roller has the obvious advantages of low cost, high efficiency, and no diameter restriction. Unfortunately, the longitudinal weld on the cathode roller adversely impacts the quality of the electrolytic copper foil due to the great difference between the microstructure of the weld zone and the base metal. Thus, it is crucial to reduce their difference by regulating the microstructure of the weld zone. In this study, a novel complex treatment of heat treatment and sandblasting is primarily developed for regulating the microstructure of the weld zone. The results show that the novel complex treatment has an efficient effect on regulating the microstructure of the weld zone and making the microstructure in the weld zone close to that of the base metal. During vacuum annealing, the microstructure of the weld zone is refined to some degree, and 650 °C annealing has the optimal effect, which can effectively reduce the ratio of α phase’s length to width and reduce the microstructure difference between the weld zone and the base metal. At the same time, with an increase in the annealing temperature, the tensile strength and yield strength decreased by about 10 MPa; the elongation after fracture increased by 20%; the average microhardness of the WZ and the HAZ decreased by about 10 HV_0.10_; and that of the BM decreased by about 3 HV_0.10_. The heat treatment after welding can effectively adjust the properties of the weld zone, reduce the hardness and strength, and improve the toughness. The subsequent sandblasting after annealing can further refine the grain size in the weld zone and make the microstructure in the weld zone close to that of the base metal. Sandblasting after annealing can further refine the grain in the weld zone and make the microstructure in the weld zone close to that of base metal. Meanwhile, an application test confirmed that the adverse impact of a longitudinal weld on the quality of electrolytic copper foil could be resolved by adopting this novel complex treatment. Therefore, this study provides valuable technical support for the “welding” manufacturing of the titanium sleeves of the cathode roller.

## 1. Introduction

Electrolytic copper foil is one of the important materials for manufacturing copper-clad laminate (CCL) and printed circuit boards (PCB) [1,2]. In recent years, due to the rapid development of science and technology in the downstream industry, higher requirements have been put forward for the upstream electrolytic copper foil. Not only is its demand increasing, but the quality requirements for copper foil are also getting higher. At present, there are two kinds of methods for producing copper foil, one is calendering, and the other is electrolysis [3,4]. Electrolytic production of copper foil is an efficient production method developed in recent years [5,6,7]. This technology has two major advantages, high efficiency and low cost [8]. It uses a roller cathode that rotates slowly at a constant speed to continuously produce a certain width of electrolytic copper foil. With the rapid development of China’s electrolytic copper foil industry, the demand for cathode rollers and the key equipment for producing copper foil is also increasing significantly yearly. As a core and key component of electrolytic copper foil equipment, the quality of the cathode roller determines the grade and quality of the copper foil.

There are two methods for manufacturing the titanium sleeve used for the cathode roller, “spinning” and “welding” [9]. The cathode roller manufactured by “welding” has many advantages, such as low production cost, high production efficiency, and it can meet flexible diameter requirements. Unfortunately, “welding” manufacturing has an unacceptable shortcoming, a longitudinal weld on the surface of the cathode roller, which forms a “bright” band on the copper foil surface in the corresponding position of the longitudinal weld and thus seriously affects the quality of the copper foil and reduces its production efficiency. Therefore, the key technology of “welding” manufacturing for a cathode roller is to develop an appropriate technical method to adjust the microstructure of the weld zone and make it close to the base metal.

It is reported that annealing treatment could refine the grain in the weld zone, but an obvious difference from the base metal still exists [10]. Therefore, it is necessary to develop a novel method to further refine the grain size in the weld zone and reduce the difference between the weld zone and base metal.

In this study, a novel complex treatment of annealing and sandblasting was primarily used to regulate the microstructure of the weld zone. The research goal is to effectively refine the grain in the weld zone and make the microstructure close to that of the base metal. 

## 2. Materials and Methods

To avoid the side effect of impurity elements, a fine-grained TA1 titanium plate with high purity and little oxygen developed by Sinosteel Precision Forging Materials Co., Ltd. was used as the raw material in this study. Its composition is shown in Table 1, and the purity of the titanium plate is very high. The titanium plate was welded by manual tungsten argon arc welding, with a welding current of 120 A and a welding voltage of 23 V. To avoid the pollution of oxygen and impurity elements in the welding wire to the weld seam, the direct welding method was adopted. The welding parameters are shown in Table 2. The welding schematic diagram and physical photograph are shown in Figure 1. The appearance of the weld seam is shown in Figure 2. It can be seen that the weld seam surface is relatively flat and smooth. After welding, the welding plate was vacuum annealed at a temperature range of 500 °C, 550 °C, 600 °C, and 650 °C for 2 h. Then, sandblasting was conducted at a pressure of 0.6 MPa with a duration of 25 min by a BA600D standard closed sandblasting machine. The DMI-3000M optical microscope was used to observe the microstructure, and the corrosion solution was a mixture solution of HF, HNO_3,_ and H_2_O, with a ratio of HF: HNO_3_: H_2_O = 2:1:40. The mechanical performance of the TA1 titanium plate at different stages was tested using the American Instron 8802 electro-hydraulic servo mechanical testing machine. The microhardness of different areas of the sample was evaluated by the HXD-1000TMC Vickers microhardness tester. Finally, an application test was conducted by preparing an electrolytic copper foil using a welded titanium plate, and the surface morphology of the electrolytic copper foil was visually observed.

## 3. Results and Discussion

### 3.1. Welded Microstructure of Titanium Plate

After the titanium plate was welded, the microstructure of the weld zone was observed and compared with that of the base metal (BM), as shown in Figure 3. It can be seen that the microstructure of the weld zone (WZ) and the base metal (BM) is greatly different. The WZ is a typical welding microstructure, mainly composed of coarsened columnar β phase along with a little acicular α phase. There are many irregular cross-flake and acicular structures due to the effect of heat input, with a much coarser grain size than the BM [11,12,13], while the microstructure in the BM is fully composed of an equiaxed α phase.

### 3.2. Annealed Microstructure of Titanium Plate

The welded plate was vacuum annealed at 500 °C, 550 °C, 600 °C, and 650 °C for 2 h, and the microstructures in the WZ after annealing at different temperatures are shown in Figure 4. It can be seen that with the increase of annealing temperature, the acicular α phase in the WZ changes in shape; that is, it changes from an acicular to a lamellar shape, so it gradually becomes similar to that of the BM. In high-temperature slow cooling, the coarse β phase in the WZ turns into a fine α phase. Therefore, the structure of the WZ is composed of α phase and a little α + β two-phase mixture, which is nearer to the structure of the BM.

### 3.3. Mechanical Performance and Microhardness of Annealed Titanium Plate

The mechanical performances of the TA1 titanium plate corresponding to each stage after annealing are shown in Table 3. It can be seen from the table that the tensile strength and yield strength of titanium plate substrate are 259 MPa and 152.5 MPa, respectively, and the elongation after fracture is 66.6%. After welding, the tensile strength and yield strength increased to 279 MPa and 183 MPa, and the elongation after fracture decreased to 20%. This may be because the titanium plate contains a small amount of impurity elements, so, during welding, these impurity elements and welds absorb oxygen and nitrogen from the air to form an interstitial solid solution, which causes a lattice distortion of the titanium, thus improving the tensile strength and yield strength, and reducing the elongation after fracture. In addition, a lot of heat is generated during welding. Thus, the surface temperature of the titanium plate increases rapidly, resulting in grain growth, and the strength of the titanium plate increases. 

After the titanium plate was welded and vacuum annealed, the tensile strength decreased from 279 MPa to 270 MPa with increased annealing temperatures. The yield strength decreased from 183 MPa to 171 MPa. The elongation after fracture increased from 20% to 40%. With an increase in annealing temperature, the tensile strength and yield strength decreased by about 10 MPa, and the elongation after fracture increased by 20%. This is because annealing causes the recrystallization of the welded titanium plate, the grain is refined to a certain extent, and the internal stress of the titanium plate is eliminated, thus reducing the tensile strength and yield strength, and improving the elongation after fracture. 

In general, annealing heat treatment helps reduce the strength after welding and improves the elongation after fracture.

Figure 5 shows the effect of annealing temperature on the microhardness of different areas of the welded plate. It can be seen from the figure that the highest hardness of the titanium plate after welding appears in the WZ, followed by the HAZ, and the lowest hardness is located in the BM. The highest hardness value appears at 2 mm from the weld seam center, about 203 HV_0.10_. The general trend is a gradual decrease from the WZ to the BM. This is because during the welding of pure titanium, due to the large amount of heat generated, the temperature at the weld seam increases rapidly, so the α phase in the WZ turns into the β phase. In the process of cooling and solidification after welding, the β phase again turns into the α phase. Still, due to the relatively fast solidification rate, only a small part of the β phase has changed. Most of the coarse β phase has been retained. Finally, the weld zone structure is a typical welding structure, mainly composed of a coarsened columnar β phase along with a little acicular α phase. At the same time, there are many irregular cross-flake and acicular structures due to the effect of heat input. Different structures in the WZ produce phase transformation strengthening, so the hardness of the WZ is on the high side. In addition, because titanium plate contains a small amount of impurity elements, during welding, these impurity elements and welds absorb oxygen and nitrogen in the air to form an interstitial solid solution, which causes lattice distortion of titanium and also increases the hardness of the WZ. The reason the highest hardness does not appear at the center of the weld seam may be because the high temperature at the center of the weld seam stays long, and some grains grow, so the hardness decreases slightly. The microstructure of the BM is completely equiaxed α phase composition, and the structure of the HAZ is affected by both the WZ and the BM. Therefore, the hardness of the HAZ takes second place, and the hardness of the BM is the lowest.

After the titanium plate is welded and then subjected to the vacuum annealing treatment, the average microhardness of the WZ and the HAZ decreases by about 10 HV_0.10_, and that of the BM decreases by about 3 HV_0.10_. In general, with the increase of annealing temperature, the microhardness of each weld zone is reduced to a certain extent. This is because annealing causes the recovery and recrystallization of the titanium plate after welding, which makes the β phase in the HAZ and the WZ turn into the α phase again. It eliminates the phase transformation strengthening caused by different structures, so the microhardness of each weld area has also been reduced to a certain extent.

In general, the peak hardness of the WZ and the HAZ decreases with the increase in annealing temperature.

### 3.4. Sandblasting Microstructure of Titanium Plate

Sandblasting is carried out after vacuum annealing, and the microstructure in the WZ before and after sandblasting is shown in Figure 6. It can be seen that the annealed microstructure of the WZ has been refined by the following sandblasting. Especially, the 650 °C annealed-sandblasted microstructure is much finer, with a grain size level close to that of the BM. 

The reason is that sandblasting provides high strain conditions for the surface layer, resulting in many dislocations on the surface layer [14,15,16,17]. High-density dislocations also gradually evolute into dislocation walls, i.e., sub-grain boundaries. As the strain increases, new dislocations are continuously generated, which evolve into new sub-grain boundaries. At the same time, the initially formed sub-grain boundaries can be evolved into grain boundaries so that the grains are refined. Therefore, the microstructure of the WZ can be further refined after sandblasting, thus reducing the difference between the microstructure of the WZ and the BM.

### 3.5. Microhardness of Each Area after Sand Blasting

After vacuum annealing, the welded plate was sandblasted. The microhardness of the corresponding weld zone under each process condition is shown in Figure 7. It can be seen that the hardness of the base material increased from about 160 HV_0.10_ to about 205 HV_0.10_ after the vacuum annealing and sandblasting of the welded plate. The maximum hardness affected by heat increased from about 180 HV_0.10_ to about 226 HV_0.10_. The maximum hardness of the weld zone increased from about 200 HV_0.10_ to about 250 HV_0.10_. In general, the microhardness of each weld area improved to a certain extent, about 45 HV_0.10_, showing that surface sandblasting effectively improves surface hardness [18,19,20]. This is because sandblasting provides high strain conditions for the surface layer of the titanium plate, resulting in a large number of dislocations on the surface layer. The dislocation density increases with the increase of deformation, and the high-density dislocation gradually forms the dislocation wall, namely the subgrain boundary; as the strain continues to increase, new dislocations are continuously generated, thus improving the microhardness of each area of the weld. This shows that surface sandblasting effectively improves surface hardness. 

### 3.6. Grain Size Grade of Weld Zone under Different Processes

The grain size scale is called grain size, usually expressed by the number of grains per unit volume (or unit area) or the average line length (or diameter) of grains. In industrial production, grain size grade is used to express grain size. The standard grain size is divided into 12 grades. Grades 1–4 are coarse grains, grades 5–8 are fine grains, and grades 9–12 are ultra-fine grains. In this paper, the grain size grade was determined by calculating the number of grains in a given region. The grain size grade of the WZ under different processes is shown in Table 4. It can be seen that the grain size grade of the BM is about 9, while the grain size grade of the WZ after welding is about 5. After vacuum annealing treatment of the welded titanium plate, the grain size grade of the weld zone after annealing at 500 °C is raised to about 6. With the annealing temperature rising from 500 °C to 650 °C, the grain size grade of the weld zone is raised from about 6 to about 7. This shows that the grain size grade of the weld zone can be effectively improved by the heat treatment of the welding retrogression. This is because annealing causes the recovery and recrystallization of the welded titanium plate, and the grain size is refined to a certain extent, so the grain size grade of the WZ is continuously improved with an increase in annealing temperature. After vacuum annealing treatment and sandblasting treatment, the grain size grade of the WZ was raised from 7 to about 8. The grain size of the WZ is about 9, only 1 grade different from the grain size after annealing + sandblasting, indicating that sandblasting can further improve the grain size grade of the WZ. This is because sandblasting provides high strain conditions for the surface layer, resulting in a large number of dislocations on the surface layer, refining the grains. This shows that the annealing + sandblasting can better refine the grain size, improve its microstructure, and reduce the difference between the WZ and the BM structure. 

### 3.7. Morphology Comparison of the Electrolytic Copper Foil

To confirm whether the adverse impact of longitudinal weld on the quality of electrolytic copper foil can be resolved by this novel complex treatment, an application test was conducted by preparing an electrolytic copper foil using a welded titanium plate with and without the novel complex treatment, and the surface morphology of the electrolytic copper foil in both cases was visually observed as shown in Figure 8. It can be clearly seen that there is an obvious bright band corresponding to the weld zone on the surface of the electrolytic copper foil without the novel complex treatment, while there is no visible bright band on the surface of the electrolytic copper foil with the novel complex treatment, indicating that the quality of electrolytic copper foil can be greatly improved by the complex treatment. In other words, the obvious bright bands disappeared, and the adverse impact of longitudinal weld on the quality of electrolytic copper foil could be resolved by adopting the novel complex treatment after welding.

## 4. Conclusions

In this study, TA1 with high purity was used as the research material, and a novel complex treatment of heat treatment and sandblasting was primarily developed for regulating the microstructure of the weld zone and making the microstructure in the weld zone close to that of the base metal. The results show that the difference between the weld area and the base metal could be effectively decreased by the novel complex treatment, and 650 °C annealing had the optimal effect of refining the microstructure of the weld zone, which could reduce the ratio of α phase’s length to width and reduce the microstructure difference between the weld zone and the base metal. At the same time, with the increase of annealing temperature, the tensile strength and yield strength decreased by about 10 MPa; the elongation after fracture increased by 20%; the average microhardness of the WZ and the HAZ decreased by about 10 HV_0.10_; and that of the BM decreased by about 3 HV_0.10_. The heat treatment after welding effectively adjusted the properties of the weld zone, reduced the hardness and strength, and improved the toughness. The subsequent sandblasting after annealing further refined the grain size in the weld zone and made the microstructure in the WZ close to that of the BM. Sandblasting increased the surface hardness by about 45 HV_0.10_. The annealing + sandblasting better refined the grain size, improved its microstructure, and reduced the difference between the WZ structure and the BM structure. Finally, the application test showed no visible bright band on the surface of the electrolytic copper foil prepared by the welded titanium plate with the novel complex treatment, which indicates that the quality of electrolytic copper foil can be greatly improved by the complex treatment. Therefore, this study can provide valuable technical support for the “welding” manufacturing of the titanium sleeve of the cathode roller.

## Figures and Tables

**Figure 1 materials-16-02149-f001:**
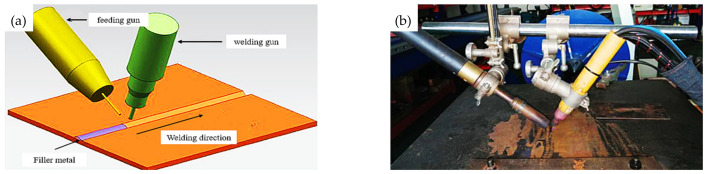
The welding schematic diagram and physical photograph. (**a**) Schematic diagram; (**b**) Physical photograph.

**Figure 2 materials-16-02149-f002:**
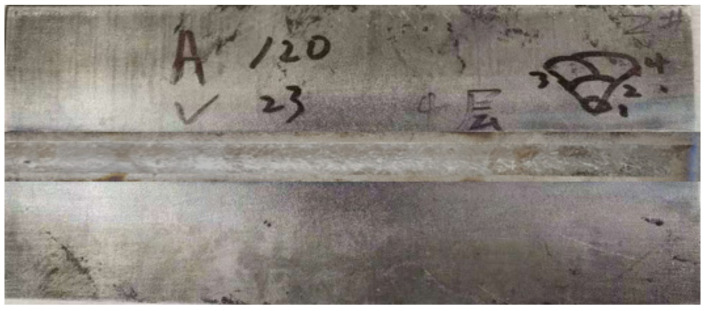
Appearance of the weld seam.

**Figure 3 materials-16-02149-f003:**
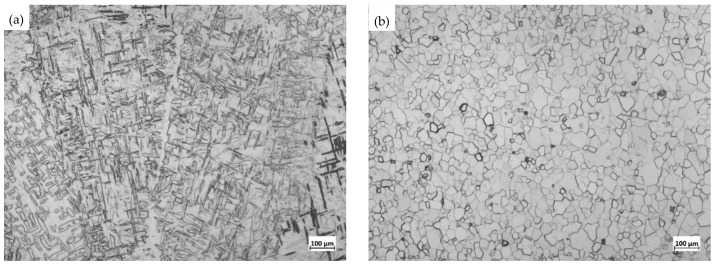
Microstructure comparison of the weld zone (WZ) and the base metal (BM). (**a**) Weld zone (WZ); (**b**) Base metal (BM).

**Figure 4 materials-16-02149-f004:**
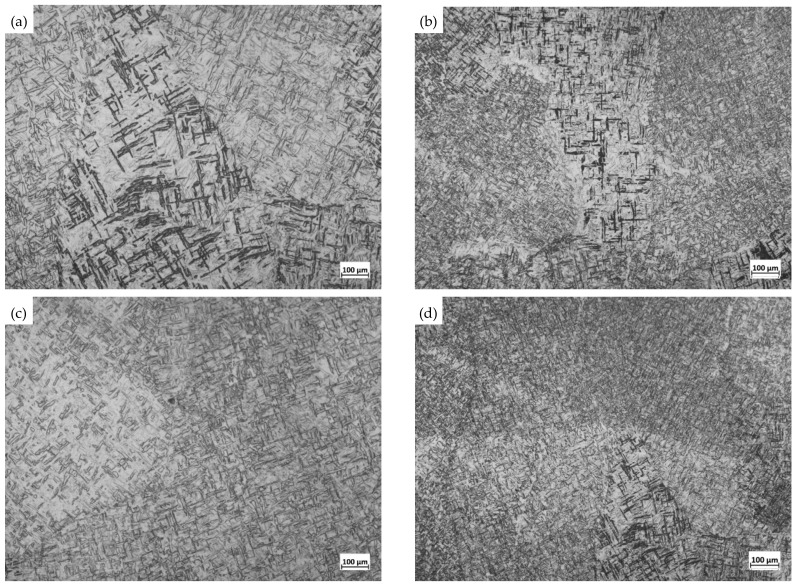
Microstructure of the weld zone (WZ) annealed at different temperatures. (**a**) 500 °C; (**b**) 550 °C; (**c**) 600 °C; (**d**) 650 °C.

**Figure 5 materials-16-02149-f005:**
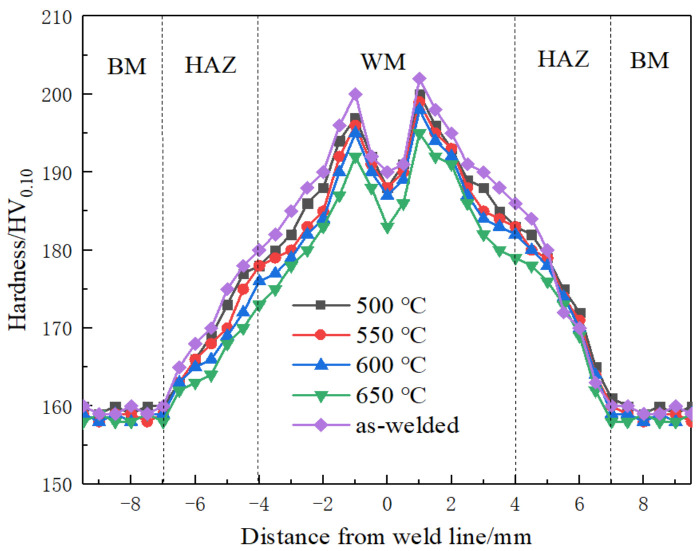
Effect of annealing temperature on microhardness of each area of welding plate.

**Figure 6 materials-16-02149-f006:**
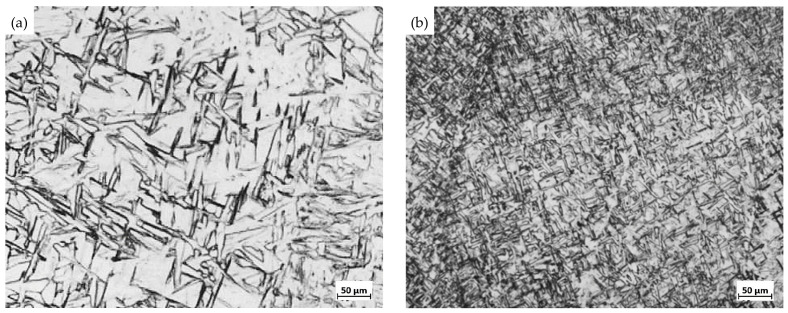
Microstructure comparison of the weld zone (WZ) before and after sandblasting annealed at different temperatures. (**a**) 500 °C; (**b**) 500 °C+ sand blasting; (**c**) 550 °C; (**d**) 550 °C + sand blasting; (**e**) 600 °C; (**f**) 600 °C+ sand blasting; (**g**) 650 °C; (**h**) 650 °C + sand blasting.

**Figure 7 materials-16-02149-f007:**
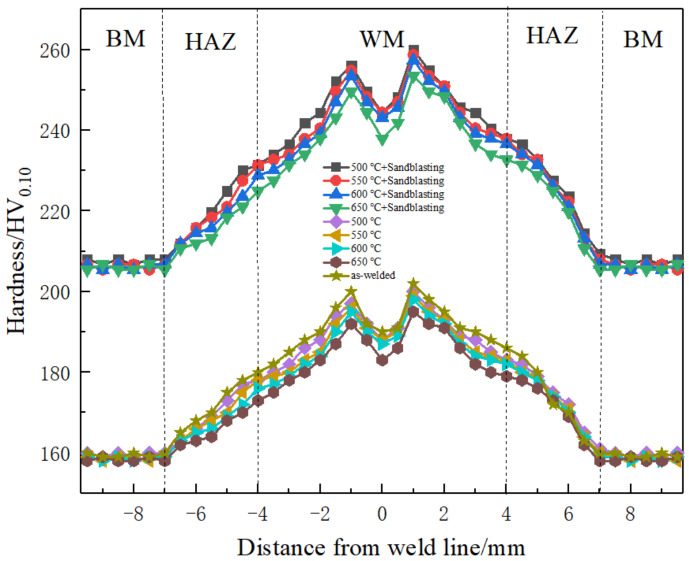
Microhardness of each area after sandblasting.

**Figure 8 materials-16-02149-f008:**
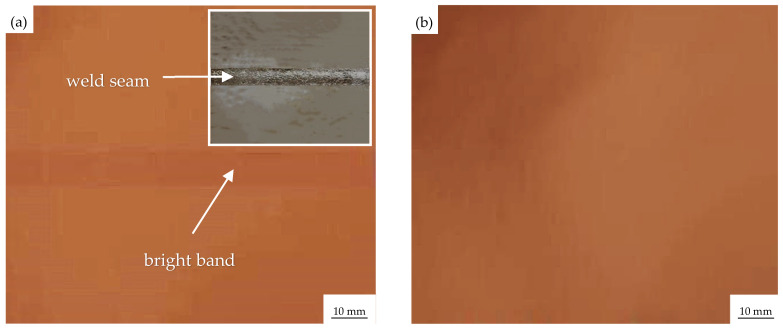
Morphology of electrolytic copper foil by welded TA1 titanium cathode roller (**a**) Without the novel complex treatment; (**b**) With the novel complex treatment.

**Table 1 materials-16-02149-t001:** Chemical composition of TA1 titanium plate (wt. %).

Elements	C	H	N	O	Fe	Ti
Content	0.007	0.0006	0.0015	0.032	0.029	Balance

**Table 2 materials-16-02149-t002:** Welding parameters of manual tungsten pole argon arc welding.

Welding Parameters	Welding Current/A	Welding Voltage/V	Welding Speed/(cm·s^−1^)	Main Nozzle/(L·min^−1^)	Support Cover/(L·min^−1^)
	120	23	0.5	20	12

**Table 3 materials-16-02149-t003:** Mechanical performance of TA1 titanium plate corresponding to each stage.

Performance		Tensile Strength/MPa	Yield Strength / MPa	Elongation after Fracture/%
Base material		259	152.5	66.6
welding		279	183	20
Annealing temperature/°C	500	277	180	35
	550	276	179	38
	600	275	175	40
	650	270	171	40

**Table 4 materials-16-02149-t004:** Grain size grade of the WZ under different processes.

Process Parameters	Original Sample	500 °C	550 °C	600 °C	650 °C	500 °C + Sandblasting	550 °C + Sandblasting	600 °C + Sandblasting	650 °C + Sandblasting
grain size grade	9	6	6.5	6.5	7	6.5	7	7.5	8

## Data Availability

The data that support the findings of this study are available from the corresponding author.

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
