# Peer review of "The Effect of Novel Complex Treatment of Annealing and Sandblasting on the Microstructure and Performance of Welded TA1 Titanium Plate"

_materials, 2023, doi:10.3390/ma16062149_

Round 1
Reviewer 1 Report
Authors used a sequence of thermal and sandblasting treatment in order to improve the welding quality of a TA1 titanium plate. Results are encouraging. The bright band of the welding was disappeared and the microstructure of the welded zone becomes close to that of the base metal.
Some statements need more clarifications. In the following, some points need to be addressed:
1) In the title and several statements of this manuscript, there is an overuse of the expression “novel complex treatment”. What is new in this study, is the sequence of treatments. The expression “complex” is not obvious. Please, clarify this complexity and use other expressions (like novel sequence of treatments) for the different statements.
2) Lines 36: “… are also increasing dramatically year by year”. Please change the expression “dramatically” by a more suitable one.
3) Line 59-60: “Then, sandblasting was conducted at a pressure of 0.6 MPa with duration of 25 min by a BA600D standard closed sandblasting machine”: What sizes of sand grains that are used in this sandblasting treatment.
4) The used scales of figure 2 and figure 3 are 100 µm and 50 µm, respectively. For the reader, to compare the final results and to observe clearly the effect of the sandblasting treatment on the microstructure, it is better to use the same scale for all photos in both figures 2 and 3. Please use the same scale for that.
Author Response
Responses to reviewer
Dear reviewer,
Thank you so much for considering our manuscript with its Ms. Ref. No. of materials-2169992 and entitled “The effect of novel complex treatment of annealing and sandblasting on regulating the microstructure of welded TA1 titanium plate”. And thank you so much for all the nice comments, which is quite valuable for improving the manuscript’s quality. We have done our best to respond all comments, as explained hereinafter. The changes to our manuscript within the document were also highlighted by using red colored text.
Comments from reviewer:
Authors used a sequence of thermal and sandblasting treatment in order to improve the welding quality of a TA1 titanium plate. Results are encouraging. The bright band of the welding was disappeared and the microstructure of the welded zone becomes close to that of the base metal.Some statements need more clarifications. In the following, some points need to be addressed:
Response:Thanks a lot for the positive comments and valuable suggestions to
improve the quality of the manuscript.
1) In the title and several statements of this manuscript, there is an overuse of the expression “novel complex treatment”. What is new in this study, is the sequence of treatments. The expression “complex” is not obvious. Please, clarify this complexity and use other expressions (like novel sequence of treatments) for the different statements.
Response: Thanks for your nice comments. The expression “complex” refers to annealing+sandblasting, which is different from the traditional single treatments, and refers to composite. Therefore, the expression “complex” not only simply summarizes the annealing+sandblasting treatments, but also more effectively shows the difference between our treatment and the traditional single treatments.
2) Lines 36: “… are also increasing dramatically year by year”. Please change the expression “dramatically” by a more suitable one.
Response: Thanks for your nice comments. We changed "dramatically" with a more appropriate expression. Please see the red mark.
3) Line 59-60: “Then, sandblasting was conducted at a pressure of 0.6 MPa with duration of 25 min by a BA600D standard closed sandblasting machine”: What sizes of sand grains that are used in this sandblasting treatment.
Response: Thanks for your nice comments. Sand grains with a diameter of 0.5 mm are used in this sandblasting treatment.
4) The used scales of figure 2 and figure 3 are 100 µm and 50 µm, respectively. For the reader, to compare the final results and to observe clearly the effect of the sandblasting treatment on the microstructure, it is better to use the same scale for all photos in both figures 2 and 3. Please use the same scale for that.
Response: Thanks for your nice comments. We have also considered using the same scale in Fig. 2 (Now it was adjusted to Fig. 4)and Fig. 3 (Now it was adjusted to Fig. 6), but the microstructure difference before and after sandblasting in Fig. 3 is not very obvious if the the same scale was used. This is because the microstructure of the weld zone has been significantly refined after annealing at 650 ℃. And it is just further refined by sandblasting, so the scale used in Fig. 2 is not suitable for Fig. 3. Therefore, after comprehensive consideration, we chose a larger magnification microscope to observe the microstructure difference before and after sandblasting, and thus the scale used in Fig. 3 is different from that used in Fig. 2.
Reviewer 2 Report
I have gone through the Paper titled “The effect of novel complex treatment of annealing and sandblasting on regulating the microstructure of welded TA1 titanium plate”.
Following were my observations.
1. The Paper has the latest literature reviews indicating the chosen topic has a present scope.
2. Line NO 57 – 60 deal with the selection of parameters carried out for the test. How did you select the parameters for welding like 120 A, and a welding voltage of 23 V?
3. How did you select the parameters for vacuum annealing temperature range of 500 ℃~650 ℃ and time as 2 hours?
4. The welding setup photographs / joint configurations/ clamping needs to be added for better clarity.
5. Figure 4 a text distorted.
6. How many samples are welded and tested under each case?
7. Is the repeatability of experiments verified?
Author Response
Responses to reviewer
Dear reviewer,
Thank you so much for considering our manuscript with its Ms. Ref. No. of materials-2169992 and entitled “The effect of novel complex treatment of annealing and sandblasting on regulating the microstructure of welded TA1 titanium plate”. And thank you so much for all the nice comments, which is quite valuable for improving the manuscript’s quality. We have done our best to respond all comments, as explained hereinafter. The changes to our manuscript within the document were also highlighted by using red colored text.
Comments from reviewer:
I have gone through the Paper titled “The effect of novel complex treatment of annealing and sandblasting on regulating the microstructure of welded TA1 titanium plate”.
1. The Paper has the latest literature reviews indicating the chosen topic has a present scope.
Response:Thanks a lot for the positive comments and valuable suggestions to
improve the quality of the manuscript.
2. Line NO 57 – 60 deal with the selection of parameters carried out for the test. How did you select the parameters for welding like 120 A, and a welding voltage of 23 V?
Response: Thanks for your nice comments. Through communication with our welding experiment teacher, the selection of welding current is mainly determined by the type of our material and the thickness of our sample. The selection of welding voltage is determined by the arc length. We have also carried out welding tests by using other welding parameters. After comprehensive consideration, the optimal parameters for welding are 120 A and 23 V.
3. How did you select the parameters for vacuum annealing temperature range of 500 ℃~650 ℃ and time as 2 hours?
Response: Thanks for your nice comments. The annealing process specification in this paper is mainly based on the fact that the recrystallization temperature of pure titanium TA1 is 500 ℃ - 600 ℃ and phase transformation temperature is 885 ℃, so the recrystallization annealing temperature should be higher than the recrystallization temperature, but lower than it α-β Phase transformation temperature, so we chose four temperatures of 500 ℃, 550 ℃, 600 ℃ and 650 ℃ for recrystallization annealing to refine the grains. The time is determined by introducing the empirical formula(t=αDK) according to the effective size (8 mm)of the sample.
4. The welding setup photographs / joint configurations/ clamping needs to be added for better clarity.
Response: Thanks for your nice comments. We have added relevant figures as shown in Figure 1.
5. Figure 4 a text distorted.
Response: Thanks for your nice comments. We adjusted Figure 4 (Now it was adjusted to Figure 8).
6. How many samples are welded and tested under each case?
Response: Thanks for your nice comments. 5 samples are used for welded and tested under each case.
7. Is the repeatability of experiments verified?
Response: Thanks for your nice comments. As answered to the last question, 5 samples are used for welded and tested under each case, and the experiments have been repeated many times, thus it is supposed to think that the repeatability of experiments is verified.
Reviewer 3 Report
1. There are two methods for manufacturing titanium sleeve used for the cathode roller, which is “spinning” and “welding”, you dealt with copper foil by welded TA1 titanium cathode roller. Dissimilar joints are very tricky , you can find a right definition in the work: Combination of friction drilling and form tapping processes on dissimilar materials for making nutless joints, Volume 232, Issue 6 https://doi.org/10.1177/0954405416661002 and how to work with data in Smart optimization of a friction-drilling process based on boosting ensembles, Journal of manufacturing systems 48, 108-121 You can see there asome of the problems you lightly presented.
2. Figure 4 is not nice, please check the size of letters.
3. Microstructure comparison of weld zone (WZ) before and after sandblasting annealed at different temperatures, Ok, but there are different processes for improving the surface state even the materials bonding, like burnishing. You can check works by A.Rodriguez about ball burnishing and rolling. A. calleja and A.Sanchez-egea worked using it in friction processes, similar to spinning, like FSW of aluminion plates.
It is reported that annealing treatment could refine the grain in the weld zone, but it still exists obvious difference from the base metal , can you give the size, the quantitative parameters of the differences?
BA600D standard closed sandblasting machine; please indicate how the machine affects resuts.
You do not define properly the sandblasting process; please define the blast grade, velocity, etc.
Electrolytic copper foil is one of the important materials for manufacturing copper 29 clad laminate (CCL) and printed circuit board (PCB), that is OK, perhaps you can include some references to alternative process, such as those using bacteria. Diaz-Tena published several works in J of Cleaner production, or in other journals. The affectation of copper using the new ideas is much better than your typical sandblasting. In the last decade, the use of high purity materials such as Oxygen-Free Copper has grown exponentially. Its use in scientific facilities, surgical equipment and high precision components defines the need for new research lines to improve processes and find more sustainable manufacturing technologies. A sustainable machining technology using a renewable natural source of tools is presented in this work. The use of bacteria as the main tools for the removal of copper has been known for many years.
Author Response
Responses to reviewer
Dear reviewer,
Thank you so much for considering our manuscript with its Ms. Ref. No. of materials-2169992 and entitled “The effect of novel complex treatment of annealing and sandblasting on regulating the microstructure of welded TA1 titanium plate”. And thank you so much for all the nice comments, which is quite valuable for improving the manuscript’s quality. We have done our best to respond all comments, as explained hereinafter. The changes to our manuscript within the document were also highlighted by using red colored text.
Comments from reviewer:
1. There are two methods for manufacturing titanium sleeve used for the cathode roller, which is “spinning” and “welding”, you dealt with copper foil by welded TA1 titanium cathode roller. Dissimilar joints are very tricky , you can find a right definition in the work: Combination of friction drilling and form tapping processes on dissimilar materials for making nutless joints, Volume 232, Issue 6 https://doi.org/10.1177/0954405416661002 and how to work with data in Smart optimization of a friction-drilling process based on boosting ensembles, Journal of manufacturing systems 48, 108-121 You can see there asome of the problems you lightly presented.
Response: Thanks for your nice comments. We have carefully read the paper you recommended.
2. Figure 4 is not nice, please check the size of letters.
Response: Thanks for your nice comments. We adjusted Figure 4(Now it was adjusted to Figure 8).
3. Microstructure comparison of weld zone (WZ) before and after sandblasting annealed at different temperatures, Ok, but there are different processes for improving the surface state even the materials bonding, like burnishing. You can check works by A.Rodriguez about ball burnishing and rolling. A. calleja and A.Sanchez-egea worked using it in friction processes, similar to spinning, like FSW of aluminion plates.
It is reported that annealing treatment could refine the grain in the weld zone, but it still exists obvious difference from the base metal , can you give the size, the quantitative parameters of the differences?
BA600D standard closed sandblasting machine; please indicate how the machine affects resuts.
You do not define properly the sandblasting process; please define the blast grade, velocity, etc.
Electrolytic copper foil is one of the important materials for manufacturing copper 29 clad laminate (CCL) and printed circuit board (PCB), that is OK, perhaps you can include some references to alternative process, such as those using bacteria. Diaz-Tena published several works in J of Cleaner production, or in other journals. The affectation of copper using the new ideas is much better than your typical sandblasting. In the last decade, the use of high purity materials such as Oxygen-Free Copper has grown exponentially. Its use in scientific facilities, surgical equipment and high precision components defines the need for new research lines to improve processes and find more sustainable manufacturing technologies. A sustainable machining technology using a renewable natural source of tools is presented in this work. The use of bacteria as the main tools for the removal of copper has been known for many years.
Response: Thanks for your nice comments. We have carefully read the paper you recommended. There are different processes for improving the surface state even the materials bonding. We just put forward another different processes to explore whether it is feasible. The experimental results show that we have achieved initial success. We have added the grain size grade and base metal grade of the weld zone at each stage to more directly show the difference between the weld zone and the base metal, please see 3.6. The sandblasting process is not the main content of our study, but as an aid. Therefore, we did not present the impact of different sandblasting processes on the experimental results, the process parameters that the best results can be obtained were chosen as the final parameters. After consulting relevant literature and our own sandblasting experiments, the blast grade, velocity can be presented by sandblasting pressure and time. Therefore, we choose the two parameters that are most easily controlled, namely pressure and time. Thank you very much for your valuable comments. After reading the literature you recommended, we found that the method in the paper can indeed be used to treat copper foil. We will carry out research in this field in the future. Now we focus on exploring the research of related processes in this paper.
Reviewer 4 Report
The manuscript "The effect of novel complex treatment of annealing and sand-blasting on regulating the microstructure of welded TA1 titanium plate" has been reviewed.
It deals with the proposal of sand blasting after annealing for the reduction of grain size in weld zone welded TA1 plate.
The manuscript is clear, well planned and arranged. My only remarks concern the adoption only of metallography for the microstructural study without support of other experimental techniques (XRD, SEM, EBSD, XPS).
In my opinion at least one of this aspect should be considered y the authors.
Author Response
Responses to reviewer
Dear reviewer,
Thank you so much for considering our manuscript with its Ms. Ref. No. of materials-2169992 and entitled “The effect of novel complex treatment of annealing and sandblasting on regulating the microstructure of welded TA1 titanium plate”. And thank you so much for all the nice comments, which is quite valuable for improving the manuscript’s quality. We have done our best to respond all comments, as explained hereinafter. The changes to our manuscript within the document were also highlighted by using red colored text.
Comments from reviewer:
The manuscript "The effect of novel complex treatment of annealing and sand-blasting on regulating the microstructure of welded TA1 titanium plate" has been reviewed.
It deals with the proposal of sand blasting after annealing for the reduction of grain size in weld zone welded TA1 plate.
The manuscript is clear, well planned and arranged. My only remarks concern the adoption only of metallography for the microstructural study without support of other experimental techniques (XRD, SEM, EBSD, XPS).
In my opinion at least one of this aspect should be considered y the authors.
Response:Thanks a lot for the positive comments and valuable suggestions to improve the quality of the manuscript. We do think that the adoption only of metallography for the microstructural study is clear enough to prove the reduction of differences between weld zone and base metal in terms of microstructure. In order to have a better understanding the effect of the treatment, we added the mechanical performance and microhardness of titanium plate at each stage, please see 3.3 and 3.5. Make this study more complete.
Round 2
Reviewer 2 Report
The authors responded all the queries properly.
Author Response
Dear reviewer,
Thank you so much for considering our manuscript with its Ms. Ref. No. of materials-2169992 and entitled “The effect of novel complex treatment of annealing and sandblasting on regulating the microstructure of welded TA1 titanium plate”. And thank you so much for all the nice comments, which is quite valuable for improving the manuscript’s quality.
Reviewer 3 Report
I thing you did not accomplish the previous review, because paper is still in the same terms. Surface treatments play a critical role in the performance and durability of coatings and welded joints. To achieve optimal results, many experts are turning to innovative techniques like ball burnishing, shot peening, and polishing. These surface treatments have been shown to significantly improve the final joining or welding union, as they help to eliminate surface imperfections, reduce residual stress, and improve the overall finish of the surface. For example, ball burnishing involves using a rotating tool with a hardened steel ball to smooth the surface and improve its finish.
Polishing is another popular surface treatment technique that involves using abrasive materials to create a smooth and shiny finish. This can be especially beneficial for coatings and welded joints, as it can help to remove any surface defects or irregularities and improve the overall appearance and performance of the surface. By using these advanced surface treatment techniques, it is possible to achieve high-quality, durable coatings and welded joints that can withstand even the most demanding environments. So, please read the previous suggestions and make the reviewer´s required changes.
Mechanical testing are perhaps a real need in this paper, tensile testing and it would be nice to discuss about a campaign for fatigue testing. Did you have some data about the longing life under complex loads?
Figure 8 b does not show anything. Perhaps another magnification would be nice.
Sandblasting after annealing is a powerful technique that can help refine the grain structure in the weld zone, leading to improved mechanical properties and enhanced performance. By removing the surface oxide layer and other impurities, sandblasting can help to expose fresh metal surfaces, which can then be annealed to create a finer grain structure. However Ball burnishing as it was defined in recent works and in many applied after FSW, for instance in Joining metrics enhancement when combining FSW and ball-burnishing in a 2050 aluminum alloy, Surface and Coatings Technology 367, 327-335
Summary: I kindly suggest that authors did not attend the previous suggestions and it would be nice another improvement round in the paper. You have only 13 references,
Author Response
Responses to reviewer
Dear reviewer,
Thank you so much for considering our manuscript with its Ms. Ref. No. of materials-2169992 and entitled “The effect of novel complex treatment of annealing and sandblasting on regulating the microstructure of welded TA1 titanium plate”. And thank you so much for all the nice comments, which is quite valuable for improving the manuscript’s quality. We have done our best to respond all comments, as explained hereinafter. The changes to our manuscript within the document were also highlighted by using red colored text.
Comments from reviewer:
I thing you did not accomplish the previous review, because paper is still in the same terms. Surface treatments play a critical role in the performance and durability of coatings and welded joints. To achieve optimal results, many experts are turning to innovative techniques like ball burnishing, shot peening, and polishing. These surface treatments have been shown to significantly improve the final joining or welding union, as they help to eliminate surface imperfections, reduce residual stress, and improve the overall finish of the surface. For example, ball burnishing involves using a rotating tool with a hardened steel ball to smooth the surface and improve its finish.
Polishing is another popular surface treatment technique that involves using abrasive materials to create a smooth and shiny finish. This can be especially beneficial for coatings and welded joints, as it can help to remove any surface defects or irregularities and improve the overall appearance and performance of the surface. By using these advanced surface treatment techniques, it is possible to achieve high-quality, durable coatings and welded joints that can withstand even the most demanding environments. So, please read the previous suggestions and make the reviewer´s required changes.
Mechanical testing are perhaps a real need in this paper, tensile testing and it would be nice to discuss about a campaign for fatigue testing. Did you have some data about the longing life under complex loads?
Figure 8 b does not show anything. Perhaps another magnification would be nice.
Sandblasting after annealing is a powerful technique that can help refine the grain structure in the weld zone, leading to improved mechanical properties and enhanced performance. By removing the surface oxide layer and other impurities, sandblasting can help to expose fresh metal surfaces, which can then be annealed to create a finer grain structure. However Ball burnishing as it was defined in recent works and in many applied after FSW, for instance in Joining metrics enhancement when combining FSW and ball-burnishing in a 2050 aluminum alloy, Surface and Coatings Technology 367, 327-335.
Summary: I kindly suggest that authors did not attend the previous suggestions and it would be nice another improvement round in the paper. You have only 13 references.
Response: Thanks for your nice comments. Figure 8 b shows the morphology of electrolytic copper foil by the cathode roll treated by the novel complex treatment after welding, it is clear that the whole surface is coincident without obvious bright band. which is the qualified appearance, and that is why you thought Figure 8 b does not show anything. Actually, having coincident appearance in the whole surface is what we need. While in Figure 8 a, you can see that there exist obvious bright band without the novel complex treatment, which will affect the quality of the copper foil. In other words, the obvious bright band disappeared by adopting the novel complex treatment, and the quality can meet the requirements.
Again, thanks for your comments about references, we read the previous suggestions once again and made the required changes after revision. And several related references were added, including those you suggested (Surface and Coatings Technology 367, 327-335; Combination of friction drilling and form tapping processes on dissimilar materials for making nutless joints, Volume 232, Issue 6; how to work with data in Smart optimization of a friction-drilling process based on boosting ensembles, Journal of manufacturing systems 48, 108-121).
Last but not least, thanks for your nice comments about fatigue testing, we will evaluate the fatigue resistace in the future work based on the real application conditions.
Reviewer 4 Report
The manuscript has been improved and can be accepted as it is.
Author Response

(The authors gave the same response as above.)
